# Clinical Applications of Liquid Biopsy in Colorectal Cancer Screening: Current Challenges and Future Perspectives

**DOI:** 10.3390/cells11213493

**Published:** 2022-11-04

**Authors:** Diana Galoș, Alecsandra Gorzo, Ovidiu Balacescu, Daniel Sur

**Affiliations:** 1Department of Medical Oncology, The Oncology Institute “Prof. Dr. Ion Chiricuţă”, 400015 Cluj-Napoca, Romania; 2Department of Genetics, Genomics and Experimental Pathology, The Oncology Institute “Prof. Dr. Ion Chiricuţă”, 400015 Cluj-Napoca, Romania; 3Department of Medical Oncology, University of Medicine and Pharmacy “Iuliu Hațieganu”, 400012 Cluj-Napoca, Romania

**Keywords:** liquid biopsy, circulating tumor cells, circulating nucleic acids, circulating DNA, microRNA, exosomes, colorectal cancer, screening

## Abstract

Colorectal cancer (CRC) represents the third most prevalent cancer worldwide and a leading cause of mortality among the population of western countries. However, CRC is frequently a preventable malignancy due to various screening tests being available. While failing to obtain real-time data, current screening methods (either endoscopic or stool-based tests) also require disagreeable preparation protocols and tissue sampling through invasive procedures, rendering adherence to CRC screening programs suboptimal. In this context, the necessity for novel, less invasive biomarkers able to identify and assess cancer at an early stage is evident. Liquid biopsy comes as a promising minimally invasive diagnostic tool, able to provide comprehensive information on tumor heterogeneity and dynamics during carcinogenesis. This review focuses on the potential use of circulating tumor cells (CTCs), circulating nucleic acids (CNAs) and extracellular vesicles as emerging liquid biopsy markers with clinical application in the setting of CRC screening. The review also examines the opportunity to implement liquid biopsy analysis during everyday practice and provides highlights on clinical trials researching blood tests designed for early cancer diagnosis. Additionally, the review explores potential applications of liquid biopsies in the era of immunotherapy.

## 1. Introduction

According to the GLOBOCAN database, colorectal cancer (CRC) is the third most prevalent cancer worldwide, representing the second leading cause of cancer-related mortality around the globe [1]. Albeit the incidence of CRC is significantly higher in western countries, lower-income communities are experiencing an increase in CRC cases [2]. Various factors have been associated with a higher risk of developing CRC, including race, older age, male sex, personal history of colorectal polyps or inflammatory bowel disease, type 2 diabetes mellitus and insulin resistance, family history of cancer of the large intestine and rectum and hereditary colorectal cancer syndromes (Lynch Syndrome, familial adenomatous polyposis, MUTYH-associated polyposis) [3,4]. Additional risk factors include obesity, increased alcohol use, smoking and frequent consumption of red and processed meat [4]. Diagnosed at an early stage (I and II) the 5-year survival rates of CRC patients reach 90% [5]. However, once the disease has spread to the lymph nodes and distant organs (stages III and IV), the survival rate decreases significantly [5,6].

Despite the high mortality rates observed in the advanced stages, CRC is frequently a preventable malignancy through screening methods [7]. The vast majority of CRC cancers progress gradually, as a result of multiple histological, morphological and genetic alterations. Before the primary tumor produces notable symptoms (pain, constipation, bowel obstruction and bleeding), the stages of CRC evolution can be observed with the help of screening tests [5]. Screening methods require either direct visualization of the lesion by endoscopic evaluation (colonoscopy, sigmoidoscopy, or computed tomography colonography), or detection of abnormal DNA and/or hemoglobin as indicators of occult blood in the stool through stool-based tests (fecal occult blood test, fecal immunochemical test, or multitarget stool DNA testing) [8].

Even with the various screening technologies available, with colonoscopy representing the gold standard, adherence to CRC screening programs remains suboptimal among some populations [9,10]. Endoscopic methods require a disagreeable bowel preparation protocol in addition to anesthesia or sedation, bearing the risks of an invasive procedure (bleeding, or bowel perforation) [5,11]. Stool-based tests have an inferior precision in detecting precancerous lesions less prone to bleeding, and their accuracy is defined by an inferior sensitivity value [11]. Therefore, a less invasive screening method, fitted to identify cancer in asymptomatic patients remains an unmet need. Liquid biopsies represent a promising novel technology, attempting to overcome the limitation of current screening technologies, as well as facilitate the staging process, assess prognosis and observe drug resistance and minimal residual disease [12].

The aim of this article is to review the existing data on liquid biopsies and their clinical application in detecting early-onset CRC. In addition, the review discusses the available methods for implementing liquid biopsies in everyday practice and offers highlights on clinical trials investigating blood tests designed for screening and early cancer detection. The review also discusses the challenges met during the analysis of liquid biopsy components and its potential future applications in the era of immunotherapy.

## 2. Liquid Biopsy

Recent research has been shedding light on a new diagnostic approach suited for cancer patients, the liquid biopsy. Liquid biopsy comes as a simple, minimally invasive diagnostic tool, attempting to overcome the limitations of conventional tissue biopsy by providing more comprehensive data on tumor heterogeneity and dynamics at different junctures in cancer development [13]. Liquid biopsy refers to the biological fluids obtained from cancer patients and submitted to extensive analysis in order to isolate biomarkers indicative of malignancy. The liquid samples considered for testing can include any biological fluid (e.g., urine, pleural effusion, ascites, sputum, or cerebrospinal fluid), however, the main focus is peripheral blood [14]. The main components of liquid biopsies are circulating tumor cells (CTCs), circulating nucleic acids (circulating tumor DNA and circulating microRNAs) and extracellular vesicles (exosomes and microvesicles) [12,15] (Figure 1). Liquid biopsies allow a comprehensive analysis of plasma, also considered the somatic component of the blood, which can be manipulated for the isolation of CTCs, circulating nucleic acids and exosomes [16]. Somatic mutations can be detected through a thorough examination of these plasma components, which are shed into the bloodstream directly from the primary tumor and distant metastasis, therefore offering an extensive characterization of the tumor mass [17]. In this context, liquid biopsy analytes are currently finding their clinical application in the setting of CRC screening.

### 2.1. Circulating Tumor Cells (CTCs)

Originally described in 1869, CTCs are now gaining clinical importance in the management of patients with cancer [18]. CTCs define cells derived from the primary tumor, metastases and recurrence sites that enter the circulatory system either as individual cells or as clusters [19]. Tumor cells constitute these clusters alone or in association with fibroblasts, leukocytes, endothelial cells and platelets, forming tumor microemboli more resistant against the aggression of the host’s immune system [20]. Once they have entered the bloodstream, CTCs bear the capacity to seed the disease to secondary sites, causing tumor metastases in distant organs and disease relapses [19] (Figure 2). Furthermore, CTCs have shown great plasticity through their ability to undergo epithelial-to-mesenchymal transition (EMT) [21]. As most CTCs entering the bloodstream are exposed to mechanical and environmental factors (oxidative stress, shear stress, immunological response, and the absence of growth factors), their clearance is particularly rapid, with a half-life usually limited to 1–2 h [19,22]. The number of CTCs varies between 1 to 10 cells per 10 mL of blood, with higher counts detected in metastatic patients compared to early-stage cancers [23,24]. Given the extremely low count of CTCs, their adequate quantification requires special enrichment, detection and characterization technologies.

CTCs can be successfully enriched through various procedures that exploit the dissimilarities observed between tumor cells and other circulating blood cells. These methods attempt to isolate CTCs based on their particular physical characteristics and their contrasting expression of cell surface proteins [25].

Positive enhancement methods, also known as label-dependent, use specific antibodies targeting molecular markers expressed by CTCs (cell-surface antigens). The membrane protein most commonly used during positive enhancement selection is the epithelial cell adhesion molecule (EpCAM), however, other cytoplasmic markers expressed by CTCs may be exploited (e.g., cytokeratin-8, -18, -19) [26]. EpCAM is a transmembrane glycoprotein overexpressed in most epithelial solid tumors (breast cancer, ovarian cancer, head and neck squamous cell cancer, as well as CRC) and it is associated with cell proliferation, migration, invasion motility and signal transduction [27]. It has been proved that circulating epithelial cells identified in cancer patients frequently carry the same genetic alterations as those observed in the primary tumor [28]. However, CTCs are described by phenotypical heterogeneity, with some CTCs failing the selection process due to a lack of marker expression [29]. In these conditions, negative selection methods could successfully isolate CTCs by identifying and excluding from analysis various non-malignant cells using antibodies recognizing cell surface markers expressed by these circulating blood cells [25,30]. In comparison, label-independent CTC enrichment methods subject CTCs to separation based on biophysical features such as density, size, deformability, electrical characteristics and invasiveness [31]. A new assay, isolation by size of tumor cells (ISET), was developed to aid the morphological, immunological and molecular description of CTCs. ISET allows CTCs isolation based on biophysical differences between cancerous cells and non-malignant blood cells, collecting tumor cells using specific filters and chemical substances [32].

Once the sample has undergone enrichment processes, CTCs require individual recognition. CTC detection can be achieved through immunocytology, molecular biology, or functional assays [33]. The most widely used approach for CTC detection facilitates direct immunological identification by using anti-EpCAM antibody-labeled ferrofluids targeting proteins expressed by CTCs, with the CTCs being further detected via fluorescence microscopy [19]. The development of this technique led to the implementation of CellSearch System as the only FDA-approved biotechnology adopted by clinical studies to detect and enhance CTCs [34]. Research has also investigated the use of diagnostic leukapheresis (DLA) as a tool for enabling CTC detection when associated with the CellSearch System. Studies have demonstrated that DLA facilitates the screening of greater blood volumes for the presence of CTCs, as CTCs have similar densities as mononuclear cells and can be extracted from the bloodstream during leukapheresis [35]. Another compelling approach to CTC detection comes in the form of the CellCollector GILUPI device, assessed as an in vivo CTC detection technique [36]. On a molecular level, CTCs can be detected by extracting nucleic acids (mRNA, DNA, miRNAs) and then further submitting them to analysis through real-time polymerase chain reaction (RT-PCR) or next-generation sequencing (NGS) [37]. Functional assays have been extensively researched as different techniques suited for CTC detection. The EPISPOT assay was tested as an in vitro CTC detection tool, able to select viable CTCs by recognizing specific proteins that are either secreted, released, or shed by cancerous cells [38]. The EPISPOT assay was tested, among other settings, in the CRC setting, with encouraging results [39]. Another approach to CTC selection comes in the form of dielectrophoresis. By utilizing specific electric fields, the DEPArray system allows the separation of single cells that can be later advanced to further comprehensive molecular characterization [40].

CTCs can be extensively characterized using a range of techniques. Profiling proteins expressed by CTCs represents one of the most commonly used methods and it requires immunostaining with antibodies to specific markers of cell proliferation and apoptosis [41]. A different approach is to characterize CTCs on a transcriptomic level by multiplex quantitative RT-PCR (qRT-PCR) analysis [42], RNA sequencing assays [43], or via in situ RNA hybridization [44]. In addition, following CTC isolation, copy-number alterations can be identified by submitting the DNA of single CTCs to whole genome sequencing (WGA) analysis [45]. Individual mutations are further determined using NGS [45]. In addition to these techniques, genomic aberrations expressed by CTCs may also be singled out via fluorescence in situ hybridization (FISH) [25].

The available scientific data showed the importance of CTCs in different CRC stages. In the nonmetastatic setting, the CTCs count was found to be lower, therefore, the cutoff for many of the clinical trials conducted was set for ≥1 CTC/7.7 mL of blood, while in the metastatic setting, the cutoff was set higher (≥5 CTCs/7.5 mL of blood) [46]. Even if the current gold standard for CRC screening and diagnosis remains colonoscopy with tissue biopsy, CTC analysis could favor better compliance among patients and a decrease in the economic burden [47].

In this regard, a prospective clinical study was presented at ASCO GI 2018 [48]. The study was conducted on 620 patients categorized as 138 healthy individuals and 438 patients with precancerous and cancerous lesions (adenomas, polyps and CRC stages I to IV). After processing blood samples from all 620 patients, CTCs were successfully captured and enumerated. The results of the study showed 88% overall accuracy for both precancerous and cancerous lesions in all stages of cancerous disease. Further research into the clinical application of liquid biopsy as a test for CRC screening has identified clusters of circulating endothelial cells derived from the tumor (ECC). ECCs were described as benign cells, originating from the tumor vasculature. By recognizing and enumerating ECCs, healthy individuals were differentiated from patients with early CRC [49]. CTCs detection may have the potential to become the new gold standard for CRC diagnostics once the specificity limitations are overcome.

Moreover, CTCs were shown to evaluate patients’ prognoses and predict metastasis and recurrence. A meta-analysis including 1847 patients from 11 studies revealed that an increased CTC baseline count represents an independent and strong prognostic factor for OS (HR = 2; 95% CI 1.5–2.7) and PFS (HR = 1.8; 95% CI 1.5–2.1) in metastatic CRC [50]. By analyzing the data from the Unicancer Prodige-14 trial, François et al. showed that a high (≥3/7.5 mL of blood) CTC count before and one month after treatment was correlated with a poor OS in metastatic CRC patients having potentially resectable liver metastasis [51]. A phase III clinical trial (VESNÚ-1) was conducted on metastatic CRC patients in the first-line setting, presenting a CTC count ≥ 3/7.5 mL blood. The study aimed to evaluate whether a four-drug chemotherapy regimen consisting of FOLFIRINOX+Bevacizumab can lead to better outcomes when compared to the three-drug regimen FOLFOX+Bevacizumab, in a high-risk population. The results showed that first-line FOLFIRINOX+Bevacizumab led to a significantly improved PFS (12 months; 95% CI 11.2–14.0) in metastatic CRC with >3 CTCs/7.5 mL blood, compared to FOLFOX+Bevacizumab (9.3 months; 95% CI 8.5–10.7) [52].

However, despite promising results reported by clinical trials, the scientific community will have to overcome several limitations in order to further use the information provided by CTCs in clinical practice.

### 2.2. Circulating Nucleic Acids (CNAs)

The analysis of circulating nucleic acids (CNAs) represents a novel minimally invasive approach, able to assess the tumor and its molecular characteristics while allowing a more accurate description of tumor heterogeneity and evolution in time. CNAs, more specifically, circulating tumor DNA (ctDNA) and circulating microRNA (miRNA), are generally isolated from blood; however, other biological fluids could represent a source of CNAs (saliva, cerebrospinal fluid, pleural effusions, urine and stool, etc.) [53]. CNAs enter the bloodstream by passive release or through active secretion. The passive release of CNAs results from increased production and subsequent shedding of cell debris produced by tumor necrosis and apoptosis [12,54,55]. The active release mechanism sees CNAs packed inside extracellular vesicles such as exosomes to be further secreted by tumor cells [56].

#### 2.2.1. Circulating Tumor DNA (ctDNA)

Numerous amounts of cell-free DNA (cfDNA) fragments are detected in blood plasma as a result of cellular death, with sizes ranging between 180 and 200 base pairs [57]. It has been proved that healthy individuals display considerably lower levels of cfDNA compared to cancer patients who experience an increased cell turnover proportionate with tumor growth [58]. Various other conditions may lead to an increase in cfDNA levels, such as infection, systemic inflammation, cerebral infarction, acute trauma, or post-transplantation; however, healthy individuals may also experience a rise in cfDNA concentrations during physical exercise [58,59]. ctDNA is a part of the cfDNA released from the tumor mass and can be identified as a double-stranded DNA fragment deriving from cfDNA. ctDNA represents a small percentage of the total cfDNA of an individual, with sizes ranging from 0.18 to 21 kilobytes [12]. Available data suggest that the quantity of ctDNA detected in cancer patients varies significantly, suggesting a correlation between cancer type, disease staging and aggressiveness, the treatment followed and its outcome [60,61]. In cancer patients, the cfDNA fraction identified as ctDNA has a particularly short circulation half-life of 16 min to several hours and carries tumor-specific somatic mutations, therefore offering a live portrayal of disease progression [60,62]. The liver represents a key player in cfDNA elimination, however, the spleen, kidneys and lymphatic circulation also participate in clearance processes. Therefore, lower levels of cfDNA may be detected in plasma regardless of organ impairment [63].

The blood samples collected require a series of manipulations in order to identify ctDNA and then further analyze the material for genetic aberrations. Firstly, samples are subjected to sequential centrifugation, isolating plasma as the main source for cfDNA [64]. ctDNA is then separated from cfDNA through specific library preparation methods [64]. Following isolation, the ctDNA fragment serves as material for subsequent analysis that can be achieved using targeted or untargeted techniques. Targeted techniques allow the identification of known, recurring genetic mutations that can further guide therapeutic decisions [64,65]. The methods currently available for targeted analysis of ctDNA are rt-PCR, digital PCR (droplet digital PCR—ddPCR and BEAMing) and targeted NGS [66,67,68]. Untargeted techniques enable the analysis of a broader part of the genome, facilitating the identification of unknown aberrations [69]. Untargeted studies can be completed using NGS assays (whole genome sequencing and whole exome sequencing) [70].

Extensive research focusing on circulating DNA (cDNA) analysis has successfully discovered its clinical utility in CRC screening. A series of markers indicative of aberrant DNA methylation have been described and utilized to detect CRC in early stages and precancerous lesions [71] (Table 1).

Tumor suppressor gene septin-9 (SEPT9) is one of the most widely researched methylation markers in CRC pathogenesis. A study conducted by Warren et al. [72] evaluated the efficiency of SEPT9 methylated DNA testing in diagnosing early cases of CRC. The test results showed promising accuracy, leading to FDA approval of the EpiProcolon assay as a CRC cancer screening test aimed at detecting SEPT9 gene methylation in DNA fragments released by cancerous cells (circulating DNA or ctDNA) [73]. The diagnostic value of the assay has been extensively reviewed through vast meta-analysis [74,75]. In recent years, more tests have investigated the value of SEPT9 methylation as an indicator of early cancer occurrence, reaching encouraging results either as an individual marker or in association with other markers [71]. Another methylation marker, SHOX2, was found to gradually increase in its levels from non-cancerous tissues to non-advanced adenomas, advanced adenomas, finally peaking in CRC cases [76]. Additional genes undergoing hypermethylation in CRC patients have been described. BCAT1 and IKZF1 were analyzed in prospective cohorts of more than 2000 patients. Following extraction from blood samples, the cfDNA was analyzed for methylation in the BCAT1 and IKZF1 genes, resulting in a strong correlation between tumor aggressiveness and positivity rates delivered through testing [77,78]. In addition, ctDNA methylation marker cg10673833 demonstrated an important accuracy value in identifying malignant tumors, as well as precancerous lesions [79]. A recently published study presented the results obtained by implementing a cfDNA methylation-based model of 11 biomarkers fit to detect advanced adenomas and early-stage CRC. The model reached a high sensitivity rate for both stage I CRC and advanced adenomas, with a strong specificity value confirmed in the validation cohort [80]. Further studies aiming to define the ability of hypermethylated cfDNA to identify CRC cancer revealed that ALX4 gene methylation could correctly confirm the presence of colon adenomas, as well as colorectal tumors. However, better screening accuracy was achieved when ALX4 was included in a panel of seven hypermethylated promoter regions (BMP3, NPTX2, RARB, SDC2, SEPT9 and VIM) [81]. Methylation of SFRP genes was observed in plasma samples collected from CRC patients and patients with colorectal adenomas. When analyzed as a panel, SFRP1 and SFRP2 methylation in association with SDC2 and PRIMA1, colorectal tumors were diagnosed with a sensitivity and specificity value greater than 90% [82]. SDC2 methylation has also been tested as an individual marker suggestive of cancerous lesions, with a sensitivity rate approaching 90% and a specificity rate above 95% [83]. SFRP2 was also investigated as an individual biomarker for CRC, although with a poorer accuracy [84]. The methylation of the SFRP1 gene was analyzed together with onscostatin M receptor (OSMR) gene, with the results showing a significant increase in expression levels in patients with colorectal adenomas and those with cancerous lesions [85].

**Table 1 cells-11-03493-t001:** Outline on methylation markers with confirmed validity in CRC screening and early diagnosis.

Gene	DNA Sample	No. of Cases	No. of Controls	Case Characteristics	Sample Type	Se. (%)	Sp. (%)	AUC Value	Observations	Ref.
SEPT9	N/A	50	94	I+II: 3/4 of all samples	Plasma	CRC: 90; I+II: 86.8;	CRC: 88;		SEPT9 DNA methylation test identifies all late-stage CRC cancers	[72]
SEPT9	ctDNA	2613	6030		Plasma Serum	48.2–95.6	79.1–99.1		Meta-analysis including 25 studies; Epi proColon 2.0 exhibits the highest diagnostic value	[74]
SEPT9	cfDNA	1801	470		Plasma	69	92		Meta-analysis including 22 studies; Epi proColon 2.0 exhibits the highest diagnostic value;	[75]
SHOX2	cfDNA	103	63	I: 7; IIA: 7; IIB: 3; IIIA: 3; IIIB: 1; IIIC: 4; IV: 3; CA: 75;	Plasma	-	-	CRC: 88 *p* < 0.001; CA: 0.90 *p* < 0.001;	SHOX2 does not distinguish CRC from CA; SHOX2 methylation levels shows gradual increase from non-cancerous lesions to CRC;	[76]
BCAT1 IZKF1	cfDNA	129	1291	I: 29; II: 42; III: 40; IV: 16; Unknown: 2; CA: 685;	Plasma	66	-		Sensitivity of BCAT1 and IZKF1 is low for CA, but increases in CRC patients according to tumor staging; Specificity of BCAT1 and IZKF1 for non-neoplastic is 94%;	[77]
BCAT1IZKF1	ctDNA	187	-	I: 40; II: 54; III: 63; IV: 30;	Plasma	62	92		BCAT1 and IZKF1 methylation levels increase with CRC stage and decrease after surgical resection;	[78]
cg10673833	ctDNA	801	1021	N/A	Plasma	89.7	86.8		Dynamic changes in cg10673833 methylation are consistent with treatment outcomes	[79]
11 methylation Markers *	cfDNA	123	67	I: 34; II: 42; III: 30; IV: 17;	Plasma	84.6	86.6	0.92	Results obtained during validation cohort	[80]
ALX4 BMP3NPTX RARB SDC2 SEPT9VIM	cfDNA	193	102	T1: 3; T2: 30; T3: 120; T4: 34; T unknown: 6; N0: 121; N1: 38; N2: 28; N unknown:6; M0: 159; M1: 34;	Plasma	CRC: 90.7; I+II: 88.7;	CRC: 72.5; I+II: 73.5;	CRC: 0.86; I+II: 0.85;	Individual hypermethylated DNA promoter regions have limited diagnostic value for CRC; The 7 hypermethylated model shows good CRC detection value;	[81]
SFRP1 SFRP2 SDC2 PRIMA1	cfDNA	84	37	CRC: 47; CA: 37;	Plasma	CRC: 91.5; CA: 89.2	CRC: 97.3; CA: 86.5		SFRP1, SFRP2, SDC2 and PRIMA1 show increased methylation in both tissue samples and plasma;	[82]
SDC2	cfDNA	131	125	I: 26; II: 57; III: 36; IV: 12;	Serum	CRC: 87	CRC: 95.2		SDC methylation concludes a 92.3% sensitivity rate for stage I CRC detection;	[83]
SFRP2	cfDNA	69	55	I: 13; II: 27; III: 17; IV: 5; AA: 7;	Serum	CRC: 69.4; AA: 42.9;	CRC: 87.3;		Diagnostic value of SFRP2 methylation for detecting CRC could improve with higher input sample volumes;	[84]
OSMR SFRP1	cfDNA	136	561	I: 38; II: 29; III: 32; IV: 15; CA: 22;	Plasma			CRC: 0.710;	Significantly higher levels of cfDNA are present in CRC patients with advanced histopathological stage;	[85]

* cg00310855, cg01857475, cg01922936, cg11320449, cg11407741, cg11596863, cg15020425, cg22329423, cg24733262, cg25300584, cg26337020; Se. (%): Sensitivity; Sp. (%): Specificity; AUC: Area Under the Curve; I: CRC stage I; II: CRC stage II; IIA: CRC stage IIA; IIB: CRC stage IIB; III: CRC stage III; IIIA: CRC stage IIIA; IIIB: CRC stage IIIB; IV: CRC stage IV; CA: colorectal adenomas; AA: advanced adenomas; T: primary tumor; N: regional lymph nodes; M: distant metastases; N/A: information not available.

Several multi-cancer detection tests have been thoroughly analyzed as potential screening tools in a new era of preventive medicine (Table 2). CancerSEEK assesses the levels of circulating protein biomarkers and ctDNA aberrations in an attempt to diagnose eight of the most common malignancies, including CRC [86]. Cohen et al. applied this test on a cohort of more than 1000 patients with previously confirmed non-metastatic cancers with impressive results [87]. PanSeer is another blood-based analysis designed to identify ctDNA methylation biomarkers [88]. The blood test was applied to 605 plasma samples obtained from asymptomatic individuals during the Taizhou Longitudinal Study [88]. Within 4 years, 191 of the controls were diagnosed with one of the five most common cancer types (lung, colorectal, stomach, esophageal, or liver cancer), with PanSeer identifying the disease in 95% of the asymptomatic controls who later developed malignancies [88]. Similarly, the Galleri test was designed to successfully identify 12 different types of cancer, including CRC, in the early stages of evolution [86] with promising results [89]. A group of European scientists proposed a series of cfDNA methylation-based panels aiming to detect early-stage colorectal neoplasia, breast cancer and lung cancer in the female population [90]. The study results proved encouraging: the PanCancer panel showed a 72% sensitivity in identifying one of the three cancers with a 74% specificity, while the employment of the CancerType panel signaled the most probable cancer localization with an 80% specificity. The ColoDefense [91] test is another blood-based CRC screening assay designed to detect methylation in the SEPT9 gene, as well as the syndecan-2 (SDC2) gene. The results of the test showed an overall sensitivity in CRC detection of nearly 90% and a specificity of 92.8%. Interestingly, of the two methylation markers, SDC2 proved a higher sensitivity in detecting advanced adenomas [91].

#### 2.2.2. Circulating microRNAs (miRNA)

Circulating miRNAs represent the purpose of extensive research focusing on cancer biomarkers. miRNA can be identified in biological fluids as a result of tissue injury and apoptosis or following active selective secretion by CTCs and tumor cells from the primary site or metastases [92,93]. The infiltrating immune cells forming the tumor microenvironment have been studied as potential origins for miRNAs [94]. miRNAs may also be found in circulation encapsulated in exosomes and microvesicles [92]. miRNA is a single-stranded, non-coding RNA molecule, of small lengths containing approximately 25 nucleotides, responsible for RNA slicing processes and post-transcriptional gene expression regulation [95]. miRNAs pair with complementary sequences of messenger RNAs (mRNAs), binding to 3′ untranslated regions (3′UTR) of target mRNAs which leads to inhibition at the translation level [96]. miRNAs represent key players in numerous biological processes, both physiological and pathological, such as immune system activation, inflammatory responses, proliferation and differentiation of cells and apoptosis [96]. Research into the role of miRNAs in oncogenesis has discovered that numerous miRNAs are either downregulated or upregulated in cancers. In addition, miRNA genes were commonly found in genomic regions linked to cancer and fragile chromosomal sites, as well as regions of heterozygosity loss. These findings have prompted the conclusion that miRNAs play an important role in tumorigenesis, acting as oncogenes, as well as tumor suppressors [97]. Following isolation from cells, tissues, or body fluids, miRNA analysis is assisted by RT-PCR, dPCR and microarrays [98,99]. qRT-PCR is the most widely used method for miRNA quantification, as it requests limited amounts of RNA while offering good sensitivity rates [100]. Microarrays allow the simultaneous detection of important numbers of circulating miRNAs [101]. In addition to these methods, NGS facilitates the analysis of both known and unknown miRNAs; however, it requires large quantities of material [55].

Even though the interest in miRNA as a liquid biopsy biomarker is relatively novel, research has already found its potential value for CRC screening (Table 3).

Several miRNA clusters appear up-regulated in CRC [102], of which the miRNA-17/92a gene cluster has been extensively researched. Members of the miRNA-17/92a cluster were found elevated in plasma samples collected from patients with CRC and precancerous lesions in multiple studies [103,104,105]. A vast meta-analysis, including more than 900 CRC patients and 638 healthy controls, proved that miRNA-17 could identify CRC cases with satisfying accuracy [106]. Furthermore, the diagnostic value of miRNA-17 was confirmed by another study set to verify the accuracy of a panel formed by eight plasma miRNAs in recognizing CR adenomas [107]. More studies found the levels of miRNA-92a to be significantly elevated in the plasma of patients with advanced adenomas and those with CRC when compared to healthy controls [108,109]. Additionally, miRNA-92a also showed prognostic value in cases of CRC [108]. The same study investigated the diagnostic value of miRNA-21 with similar results [108]. More members of the miRNA-17/92a cluster, notably miRNA-19a and miRNA-19b, were found up-regulated in patients with CRC and those with precancerous lesions [104]. miRNA-18a, another member of the miRNA-17/92a cluster, demonstrated an 84.6% sensitivity and 75.6% specificity in diagnosing CRC when tested in association with miRNA-200c [110]. miRNA-18a also showed significant upregulation in patients with advanced adenomas [104,111]. Further studies found miRNA-29a to have an important diagnostic value for advanced adenomas and CRC cases when analyzed together with miRNA-92a [109]. miRNA-29a was further investigated in the setting of CRC and premalignant polyps, showing significant upregulation and valuable potential as a non-invasive biomarker for CRC screening [104]. A recent meta-analysis initiated in China sought to assess the role of circulating miRNA-21 in the diagnosis of CRC. The study was conducted on approximately 2000 samples collected from CRC patients, as well as healthy individuals, resulting in a sensitivity and specificity greater than 75% for CRC diagnosis [112]. Interestingly, a separate study found miRNA-21 levels to be elevated several years before diagnosis [113].

**Table 3 cells-11-03493-t003:** Outline on blood circulating microRNAs with confirmed validity in CRC screening and early diagnosis.

Circulating miRNA	No. of Cases	No. of Control	Case Characteristics	Sample Type	Expression	Se. (%)	Sp. (%)	Observations	Ref.
miR-92	25	20	N/A	Plasma	↑	89	70	Plasma levels reduced after surgery in 10 patients	[103]
miR-18a	123	73	I: 12; II: 21; III: 22; IV: 8; AA: 60;	Plasma	↑	-	-	miR-18a shows upregulation in AA vs. controls	[104]
miR-19a									
miR-19b									
miR-15b									
miR-29a									
miR-335									
miR-19a+ mir-19b						78.5	92.4		
miR-19a+ miR-19b+ miR-15b						78.5	79.2		
miR-19a+ miR-19b+ miR-15b+ miR-29a+ miR-335+ miR-18a	197	100	I: 20; II: 23; III: 34; IV: 14; Unknown: 5; AA: 101;	Plasma	↑	CRC: 91; AA: 95;	CRC: 90; AA: 90;	Detection rate in early CRC shows comparable results with late CRC	[105]
miR-17	938	638		Serum Plasma Stool	↑	75	68	Meta-analysis including 10 studies	[106]
miR-532-3p+ miR-331+ miR-195+ miR-17+ miR-142-3p+ miR-15b+ miR-532+ miR-652	61	26	I: 3; II: 12; III: 15; IV: 15; AA: 16;	Plasma	↑	AA: 88;	AA: 64;	Average polyp size in the validation group: 1.6 cm;	[107]
miR-431+ miR-139-3p						CRC: 91;	CRC: 57;		
miR-21+ miR-92a	250	80	CRC: 200; AA: 50;	Serum	↑	CRC: 68; AA: 70;	CRC: 91.2; AA: 70;	Overexpression of miR-92a is independently associated with poor survival	[108]
miR-29a	137	59	I: 27; II: 25; III: 38; IV: 10; AA: 37;	Plasma	↑	CRC: 69	CRC: 89.1	AA express lower levels of miR-29a and miR-92a compared to CRC miR-29a can act as both tumor suppressor and oncogene.	[109]
						AA: 62.2	AA: 84.7		
miR-92a						CRC: 84	CRC: 71.2		
						AA: 64.9	AA: 81.4		
miR-29a+ miR-92a						CRC: 83	CRC: 84.7		
						AA: 73	AA: 79.7		
miR-200c	78	86	I, II: 36; III, IV: 42;	Plasma	↑	64.1	73.3	Plasma levels of miR-18a show a tendency to increase with TNM stage	[110]
miR-18a						73.1	79.1		
miR-18a+ miR-200c						84.6	75.6		
miR-18a	66	24	CRC: 30; IBD: 18; CP: 18;	Serum	↑	-	-	Of the significantly ↑miR in CR diseases, only miR-18 shows significant↑ in CP;	[111]
miR-223	100	N/A	N/A					Of the significantly ↑miRs in CRC, only miR-223 shows significant ↑ in the validation set;	
miR-21	1129	951	N/A	Plasma Serum Stool	↑	77	83	Meta-analysis including 18 studies	[112]
miR-18a+ miR-21+ miR-22+ miR-25	77	134	I: 10; II: 21; III: 15; IV: 21;	Plasma	↑	67	90	Serum miR-21 appear elevated several years before diagnosis	[113]

Se. (%) Sensitivity; Sp. (%): Specificity; I: CRC stage I; II: CRC stage II; IIA: CRC stage IIA; IIB: CRC stage IIB; III: CRC stage III; IIIA: CRC stage IIIA; IIIB: CRC stage IIIB; IV: CRC stage IV; N/A: information not available; AA: advanced adenomas; CP: colorectal polyps; IBD: inflammatory bowel disease; ↑: upregulated expression.

### 2.3. Exosomes

Present in all biological fluids, exosomes represent cell-derived nanovesicles with sizes ranging from 30 to 150 nm in diameter [114]. Exosomes develop from the intracellular endosomal compartment following a process of inward expansion from the limiting membrane that generates multivesicular bodies (MVBs) [115]. MVBs are then discharged into the extracellular matrix as a result of their fusion with the cytoplasmic membrane, releasing their content in the form of exosomes [116]. Exosomal secretion occurs in both physiological and pathological processes, with various types of cells producing exosomes (cancer cells, as well as adipocytes, immune cells and brain cells) [117]. Exosomes play a series of roles within the cell, notably the removal of waste, antigen presentation and cytokine release [118]. However, their essential function is intercell communication of molecular information [119]. Exosomal cargo, mainly cytoplasmic components such as proteins, bio-functional lipids and nucleic acids (mRNA, miRNA and DNA fragments) are key players in the signaling pathways between cells [120]. Communication between exosomes and their target cells can be acquired through interaction with surface-expressed ligands, through phagocytosis, or by exosomal fusion with cell membranes [121].

An increasing amount of evidence has identified exosomes as essential participants in cancer development processes. Exosomes secreted by tumor cells have been found responsible for alterations in the immune response that lead to suppression of antitumor response [122]. In addition, exosomes play important roles in tumor microenvironment (TME) remodeling, angiogenesis and tumor growth, therefore favoring disease progression [118]. Exosomes were found to promote EMT, migration and invasion, causing distant cancer dissemination through various proteins and miRNAs [123]. Furthermore, research has proved that exosomal signaling pathways are also involved in therapy resistance [123,124]. Since exosomes have the ability to target specific cells [125], technologies have investigated the possibility of manipulating exosomes for therapeutical purposes, by using them as potential vehicles for drug delivery inside tumor cells [126].

Exosome concentration in biological fluids is relatively low, thus making their isolation, detection and further analysis relatively challenging. Isolation of exosomes can be achieved based on their physical properties (size and density), electromagnetic characteristics, or according to their immunological properties [127]. Isolation through ultracentrifugation represents the gold standard, however, other techniques, such as ultrafiltration, chromatography, hydrostatic filtration dialysis, precipitation, microfluidic chips, or immunoaffinity-based methods, may also facilitate exosome isolation [127]. Following isolation, the detection and characterization of exosomes from a morphological point of view is obtained through transmission electron microscopy [128]. Nanoparticle tracking analysis can determine size and concentration characteristics [129]. Protein expression and functions can be assessed using Western blot and ELISA methods, while exosome content can be identified via spectrophotometric assays and different other focused approaches (e.g., RT-qPCR, Western blot and mass spectrometry) [127,130].

Exosomal microRNAs have been attracting considerable attention as promising biomarkers suitable for cancer diagnosis. Multiple exosomal miRNAs have been studied, with some of them showing substantial value in identifying early cases of CRC (Table 4).

Research has identified several significantly up-regulated miRNAs in the setting of CRC, including cases of early-stage disease. One study found miRNA-23a to have an important diagnostic accuracy with an area under the curve (AUC) of 0.953, while miRNA-1246 and miRNA-21 also showed encouraging results [131]. Overexpression of exosomal miRNA-23a was successfully confirmed by another study that additionally investigated the diagnostic value of exosomal miRNA-301a [132]. Another study conducted by a Chinese team demonstrated the overexpression of exosomal miRNA-6803-5p in the serum of CRC patients [133]. Similarly, the expression of miRNA-486-5p in plasma exosomes was found to be significantly upregulated in CRC patients, corresponding to disease staging [134]. Circulating exosomal miRNA-125a-3p is another upregulated analyte with confirmed value in detecting early-stage CRC patients. The predictive accuracy was improved when associating miRNA-125a-3p with CEA analysis [135]. The same study found exosomal miRNA-320c to be up-regulated in plasma samples collected from CRC patients, validating it as a potential biomarker for early diagnosis [135]. Downregulation of serum exosomal miRNA-150-5p proved to be a conclusive indicator of colorectal malignancy, while combined analysis with CEA resulted in a higher diagnostic value for CRC identification [136]. Interestingly, expression of exosomal miRNA-92b was found to be significantly reduced in patients presenting with CRC, as well as individuals with adenomas of the colon [137]. Furthermore, serum samples and serum exosomes isolated from CRC patients also showed a considerable expression of mRNA-196b-5p [138]. Additional data identified a downregulation in exosomal miRNA-139-3p expression in plasma samples collected from patients with CRC, correlating with disease aggressiveness [139]. Furthermore, plasma collected from CRC patients was found to express important levels of exosomal miRNA-27a and miRNA-130a [140]. A different study found upregulation of miRNA-1359 in exosomes isolated from CRC patients, leading to an accurate differentiation of CRC cases from healthy individuals [141]. Exosomal miRNAs are receiving a growing interest, with several other studies aiming to identify and validate promising biomarkers for early CRC diagnosis and population screening [131,142,143,144]. In the given circumstances, the need to improve the accuracy of these biomarkers is evident.

## 3. The Future Applications of Liquid Biopsies in CRC

Over the past few years, immunotherapy has changed the treatment paradigm for many cancer types. In CRC, the MSI-high phenotype was associated with a significant response to immune checkpoint inhibitors (ICIs) [145]. Moreover, the tumor mutational burden (TMB), referring to the number of somatic mutations, was significantly correlated with the outcome of CRC patients treated with ICIs [146]. Despite the promising results reported by the scientific community, we face a poor prediction of response to ICIs, along with important rates of innate or acquired resistance leading to heterogenous responses among patients [147]. Biomarker-directed use immunotherapy is an important frontier in precision medicine.

To date, liquid biopsies are investigated for use as biomarkers to predict and evaluate the response to immunotherapy. CTCs, circulating DNA (cDNA), circulating RNA (cRNA) and exosomes hold a generous amount of tumor-related information. Moreover, liquid biopsies may provide a more comprehensive and dynamic overview of the tumor microenvironment and heterogeneity than single-site tissue biopsies [148]. The utility of cDNA as a prognostic and predictive biomarker for immunotherapy was shown in a phase II trial including patients with advanced or metastatic solid tumors treated with an anti-PD1 agent. The study reported that higher pretreatment variant allele frequencies (VAF) were associated with a poorer OS. However, on-treatment VAF and on-treatment reduction in VAF were correlated with longer PFS and OS [149]. These findings suggest that on-treatment cDNA variations can predict a beneficial response to ICIs. Similarly, another phase II prospective trial assessed cDNA in patients with advanced solid tumors under treatment with pembrolizumab. Low baseline cDNA levels were correlated with PFS, OS, clinical response and clinical benefit. Moreover, the reduction of cDNA after only two cycles of pembrolizumab and cDNA clearance on-treatment identified a good prognosis subset of patients [150].

A better selection of the patients receiving immunotherapy could be guided by specific somatic mutations detectable in cDNA. In this regard, a genomic mutation signature was developed to characterize immunophenotypes and predict response to immunotherapy in gastrointestinal cancers [151]. As mentioned above, the number of somatic mutations known as TMB represents an independent predictor of response to ICIs in many solid tumors, including CRC. TMB-high cases (≥20 mutations/megabase) typically occur in microsatellite instable tumors (MSI) or those harboring pathogenic mutations emerging in the DNA polymerases POLD and POLE [152]. Currently, the standard evaluation for TMB is based on tissue samples and encounters many limitations. Tissue-based biopsies cannot correctly assess intratumoral heterogeneity or evaluate the changes occurring during treatment [153]. In this regard, cDNA-based evaluation of TMB (cTMB) is currently being investigated in clinical trials, with encouraging results obtained in non-small cell lung cancer (NSCLC). The concordance between tissue-based TMB and cDNA-based TMB was strong in clinical trials, suggesting that cTMB could be a feasible predictive biomarker for ICIs [154]. Due to recent technological advances, circulating tumor cell PD-L1 expression is being investigated in clinical trials as a predictive biomarker for response to ICIs [155]. However, further scientific evidence is needed to clarify the similarities between PD-L1 detection on CTCs and tissue expression of PD-L1. Moreover, considering their rarity in the bloodstream, the utility of CTCs in immunotherapy is still in the early stages [156]. Nonetheless, emerging studies are documenting the role of extravesicles (EVs) as potential biomarkers for immunotherapy. Therefore, EV-based liquid biopsies could eventually identify tumor-expressed proteins, DNA mutations, RNA landscape, and T-cell reactivity in patients under treatment with ICIs [157].

RAS assessment in mCRC is essential to select patients suitable for anti-EGFR therapy. The concordance between tissue detection and somatic mutations detected in ctDNA appeared high in patients with advanced tumors, supporting blood-based testing. Moreover, ctDNA was shown to be highly useful for monitoring treatment response. However, some clinicopathological features, including tumor histology (mucinous) and metastatic sites (peritoneal, lung), negatively influenced RAS detection in ctDNA [158]. Along with RAS mutation, TP53 mutations were widely detected in the ctDNA of CRC patients, with a high correlation between tissue and plasma detection. In CRC patients who did not progress to metastatic disease after primary surgery, the VAF for TP53 mutations decreased. By contrast, increased levels were associated with the development of liver metastasis [159]. Moreover, TP53 mutations were significantly correlated with increased VEGFA mRNA tissue expression, suggesting that these patients are expected to benefit from anti-VEGF therapy [160]. Nonetheless, TP53 mutations might occur as a consequence of several treatment strategies. In CRC, these mutations were particularly linked to cetuximab therapy, leading to resistant clones, and, therefore, influencing treatment opportunities [161].

Immunotherapy and targeted therapy are major therapeutic breakthroughs in cancer care, and one of the most challenging concerns is proper patient selection. To overcome these shortcomings, liquid biopsy-based biomarkers represent a promising tool, hence they require detection methods with sufficient specificity, sensitivity and predictive power [162].

## 4. Remaining Obstacles in Clinical Applications of Liquid Biopsy

Despite many pieces of scientific evidence highlighting the potential benefits of liquid biopsies in cancer care, numerous limitations remain for their clinical use.

CTCs have great potential as diagnostic, prognostic, predictive, as well as monitoring tools. However, their translation into clinical practice is still restricted amid their isolation from the bloodstream [163]. To correctly identify and analyze CTCs, it is essential to understand the obstacles surrounding their use. An important challenge is their extreme rarity, which makes them hard to locate. Among blood cells, CTCs are considered to be one in a million, or a billion [164]. Moreover, their concentration is much lower in the early stages compared to metastatic disease. Another important issue regards the size, physical characteristics and complexity of surface protein expression [165]. The main techniques used for CTCs enrichment are antigen-dependent (positive or negative selection), antigen-independent, or a combination of both [166]. In the case of positive selection, the efficacy of CTCs affinity is mainly influenced by antibody selectivity in the enrichment process. In this regard, the antibodies utilized may suffer from low selectivity as the target cell-surface proteins could also be expressed on other cells. Antibody cocktails targeting several cell-surface proteins have been used to overcome this limitation [166]. Moreover, the negative enrichment methods using magnetic bands to deplete circulating platelets and leucocytes have also been used to overcome the limitations of positive enrichment [167]. While using antigen-independent methods, the isolation of CTCs depends on the electric charge, density, size and deformability. However, with these techniques, CTCs purity is usually low due to size overlapping with WBC [168].

For exosomal segregation, ultracentrifugation is thought to be the most efficient method. Although convenient and requiring reasonable costs, ultracentrifugation encounters several limitations. A significant concern is the co-purification of lipoproteins and protein aggregates along with EVs [169]. Hence, it could be overcome by combining ultracentrifugation with density-gradient mechanisms. However, the structures with similar densities are indistinguishable. Immune isolation targets EVs with a particular surface marker representing a more specific method. Nonetheless, the targeted surface marker could also be found on other EV subsets [170].

ctDNA is a small fraction representing about 0.01% of cfDNA. One of the main logistical reasons limiting the extensive use of cDNA-based analysis is represented by their feasibility outside the academic cancer centers [171]. The available techniques for cDNA detection are based on PCR and NGS. These techniques were updated over time to better fit the low concentrations in the bloodstream. However, despite their sensitivity, PCR-based assays are limited by a low multiplexing capacity that permits the analysis of a small number of loci [172]. On the other hand, the sensitivity of NGS-based assays is low and inversely proportional to the examined loci [163,173]. Another concerning issue implies the predictive value of a small set of mutations which could also be found in healthy individuals due to clonal hematopoiesis [173]. Moreover, the preanalytical sample preparation of cDNA lacks standardization limiting its implementation into clinical practice. Currently, a significant limitation preventing cDNA molecular panels in CRC is the lack of precise data showing that liquid biopsy findings could drive the therapeutic approach [174]. Similar to cDNA, the most relevant limitation in cRNA analysis is represented by preanalytical and analytical phases, along with a lack of standard extraction protocols [175]. A critical issue implies the hemolytic process, which occurs during extraction and preparation and can influence the levels of detected miRNAs. For this reason, monitoring the hemolysis of all samples in a pre-analytical phase is mandatory [176]. Another essential constraint regards residual platelets and microparticles resulting from plasma processing that can influence miRNA measurements [177]. Moreover, it is difficult to determine if the levels of plasma miRNA are confounded by comorbidities or are cancer-related. Therefore, a significant challenge is establishing which body fluid detection method is the most appropriate for CRC screening [78].

Even if several logistic and biological limitations are encountered at the present moment, liquid biopsies will more likely become a fundamental tool in the management of CRC patients in all stages of disease-related interventions. Evaluation of ctDNA levels in patients with stage II CRC has led to a decrease in adjuvant chemotherapy administration while maintaining a favorable recurrence-free survival (DYNAMIC II study) [178]. Additionally, the ongoing DYNAMIC III study attempts to evaluate a ctDNA-guided treatment approach in the setting of stage III CRC [179]. Liquid biopsy also plays an important role in assessing minimal residual disease, guiding the timing of therapeutic interventions based on ctDNA levels obtained at crucial points during treatment [180]. Furthermore, extensive research has proved that liquid biopsy offers strong indicators of early disease recurrence, with a significant median lead time of 8.7 months over conventional assessment methods [181]. Liquid biopsies have demonstrated valuable clinical application in detecting patients with acquired resistance to anti-EGFR therapy, therefore allowing a better selection of cases who may benefit from an EGFR rechallenge [182]. For CRC early diagnosis, however, comprehensive studies comparing the available data on liquid biopsy to current screening methods are necessary in order to find the most advantageous clinical use in everyday practice.

## 5. Conclusions

Despite the implementation of extensive programs designed to diagnose CRC in the early stages, the adherence of the general population to screening protocols remains unsatisfactory. Liquid biopsy comes as a novel, minimally invasive tool, adequate for early diagnosis of malignancy while also attempting to overcome the limitation showed by traditional CRC screening and tissue sampling methods. Research has proved that liquid biopsy analytes and biomarkers offer valuable information regarding carcinogenesis and could indeed single out individuals at risk for developing CRC or those presenting early-stage CRC otherwise undetected during conventional screening tests.

Since FDA has taken the necessary steps to approve CellSearch System for CTC detection and enhancement during clinical studies, the scientific community could soon implement the test in everyday clinical practice. In addition, studies have identified numerous methylation markers indicative of malignancy. In this regard, the detection of SEPT9 gene methylation in DNA fragments derivative from tumor masses is the most widely explored sequence and therefore experiences more promising results. As a result of thorough research, the SEPT9 methylation model comes within reach of validation for current practice, with tests such as Epi proColon expressing significant diagnostic values for CRC early detection. Several extensively researched miRNAs are also approaching clinical utility in everyday practice. miRNA-92a proves relevant sensitivity and specificity values in detecting early CRC cases, as well as advanced adenomas, while panels including multiple miRNAs show improved accuracy when compared to single markers. Additionally, exosomal miRNA-23a has been shown to differentiate patients with early-stage disease with high accuracy, indicating that protocols defining its clinical applicability could be imminent.

However, in order to successfully isolate compelling biomarkers, blood samples require complex manipulation techniques that often show unsatisfactory efficiency and prove to be time-consuming and costly. Studies conducted thus far have demonstrated encouraging accuracy, yet further research is critical in order to validate liquid biopsy as a screening and early diagnostic technique. Furthermore, the utility of liquid biopsies in the era of precision medicine is an important frontier that demands thorough inquiry.

## Figures and Tables

**Figure 1 cells-11-03493-f001:**
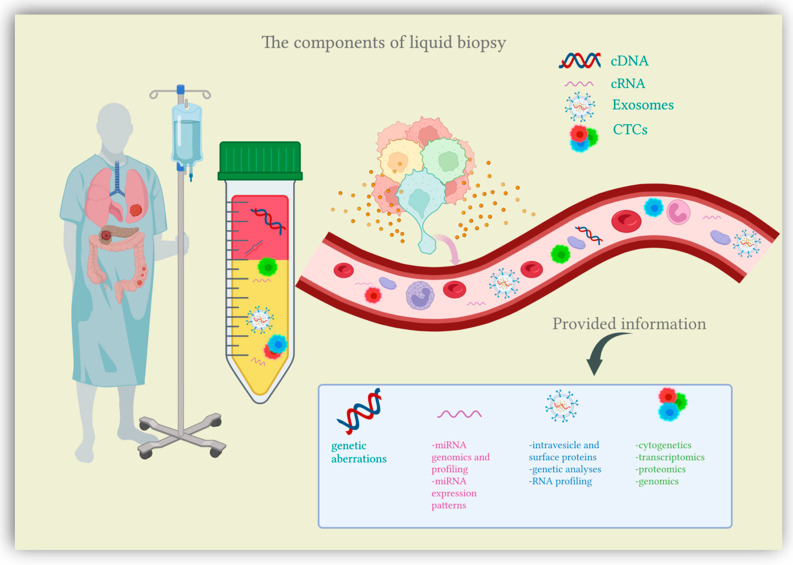
The components of liquid biopsy. Circulating DNA (cDNA), circulating RNA (cRNA), exosomes and circulating tumor cells (CTCs) are shed into the bloodstream directly from the tumor mass or metastases. Following blood sample collection, these components are further analyzed to provide extensive tumoral characterization.

**Figure 2 cells-11-03493-f002:**
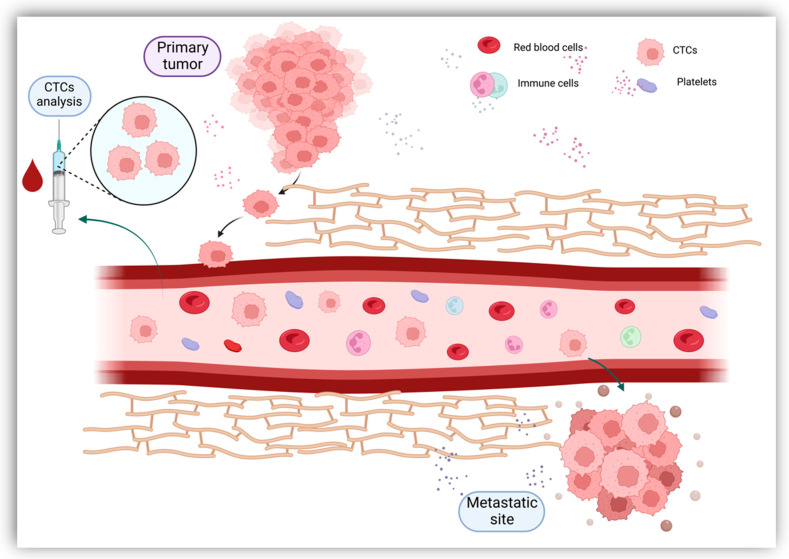
CTCs mechanism of disease dissemination. CTCs derive from the primary tumor, metastases and recurrence sites. Once they have entered the bloodstream, CTCs can metastasize in secondary sites. By collecting blood samples, CTCs are examined to offer an in-depth description of the disease.

**Table 2 cells-11-03493-t002:** Outline on blood-based liquid biopsy tests with confirmed validity in CRC screening and early diagnosis.

Test	Analytes	Purpose of Test	Case Characteristics	Observations	Ref.
CancerSEEK	ctDNA; Protein biomarkers released by tumors;	Detection of multiple types of cancer;	1005 patients with stage I to III cancers of the breast, colorectum, esophagus, liver, lung, ovary, pancreas, stomach;	The highest prediction accuracy appears for CRC; The test shows a specificity >99% in detecting the 8 types of cancer; Sensitivity rates vary from 33% (breast cancer) to 98% (ovarian cancer); The median sensitivity varies from 48% (stage I) to 78% (stage III);	[87]
PanSeer	cfDNA	Detection of cancer in asymptomatic individuals;	Plasma samples collected from 123,115 healthy subjects who were monitored over 10 years for cancer detection	The tests show 96% specificity; The test detects cancer in 95% of asymptomatic individuals who are later diagnosed with one of 5 cancers (stomach, colorectal, liver, lung, esophagus).	[88]
Galleri	cfDNA	Detection of distinct methylation patterns associated with specific cancers; Provide information about the organ of origin;	Plasma samples collected from 15,254 participants (44%—non cancer patients; 56%—cancer patients) with 50+ cancer types;	The test detects 12 types of cancer in early stages (anorectal, colorectal, esophageal, gastric, head and neck, HR+ breast, liver, lung, ovarian, pancreatic, MM, lymphoid neoplasms); The test sets a 99.3% specificity; Identification of tissue of cancer origin shows a 93% accuracy; Detection rate increases with tumor stage;	[86,89]
PanCancer and CanceType	cfDNA	Simultaneous detection of breast cancer, CRC and lung cancer based on a cfDNA methylation model;	Plasma samples collected from female patients with breast cancer, CRC and lung cancer, as well as asymptomatic controls;	PanCancer panel detects cancer cases with a 72.4% sensitivity and 73.5% specificity; CancerType panel indicates the most likely cancer topography with specificity of over 80%, but with limited sensitivity;	[90]
ColoDefense	cfDNA	Combined detection of SEPT9 and SDC2 methylation for improved detection of AA and early-stage CRC;	Plasma samples collected from 117 CRC patients, 23 patients with AA, 78 patients with small polyps and 166 normal individuals;	CRC detection shows an overall sensitivity of 88.9% and specificity of 92.8%; Test results prove a significantly improved accuracy compared to the single methylation marker detection;	[91]

HR+: hormone receptor-positive; MM: multiple myeloma; AA: advanced adenomas.

**Table 4 cells-11-03493-t004:** Outline of blood exosomal molecules with confirmed validity in CRC screening and early diagnosis.

Exosomal miRNA	No. of Cases	No. of Control	Case Characteristic	Sample Type	Expression	AUC Value	*p* Value	Observations	Ref.
miR-23a	88	11	I: 20; II: 20; IIIA: 20; IIIB: 16; IV: 12;	Serum	↑	0.953	<0.0001	Exosomal levels of miRNAs are not dependent on clinical CRC stage Exosomal levels of miR-23a and miR-1246 prove better sensitivity for stage I CRC detection than CEA and CA19-9 assessment	[131]
miR-1246						0.948	<0.001		
miR-21						0.798	<0.0001		
miR-150						0.758	<0.0001		
let-7a						0.670	<0.0001		
miR-223						0.716	<0.0001		
miR-1224-5p						0.610	=0.142		
miR-1229						0.614	<0.0001		
miR-23a	25	13	II: 12; III: 13;	Serum	↑	0.890	<0.05	Exosomal levels of miRNAs are not correlated with clinicopathological characteristics of CRC cases	[132]
miR-301a						0.840	<0.05		
miR-6803-5p	168	20	I: 21; II: 48; III: 68; IV: 31;	Serum	↑	0.740	<0.05	High levels of exosomal miR-6803-5p correlates with advanced TNM stage, lymph node metastases, liver metastases, poorer DFS and OS	[133]
miR-486-5p	50	50	I+II: 25; III+IV: 25;	Plasma	↑	0.713	<0.05	High expression of exosomal mR-486-5p in CRC samples contrasts low expression of miR-468-5p in CRC tissue samples	[134]
miR-125a-3p	50	50	I: 3; IIA: 43; IIB: 4;	Plasma	↑	0.685	<0.001	Exosome miR-125a-3p and miR-320c levels correlate with tumoral nerve infiltration	[135]
miR-125a-3p+CEA						0.855	<0.0001		
miR-320c						0.598	=0.145		
miR-150-5p	133	60	I: 32; II: 43; III: 28; IV: 30;	Serum	↓	0.870	<0.05	Decreased exosomal miR-150-3p correlates with advanced TNM stage, lymph node metastases, poorly differentiated tumors	[136]
miR-150-5p+CEA						0.910	<0.05		
miR-92b	62	52	I: 22; II: 9; III: 6; Unknown: 3; CA: 22;	Plasma	↓	0.793	<0.001	Highest accuracy reported for differentiating CRC II/III from NC	[137]
miR-196b-5p	150	90	N/A	Serum	↑	0.880	<0.001	Exosomal miR-196b-5p detects CRC with higher accuracy than serum miR-196b-5p	[138]
miR-139-3p	80	23	T1+T2: 26; T3+T4: 54; N0: 42; N1: 18; N2: 20; M0: 78; M1: 2;	Plasma	↓	0.726	<0.001	Levels correlate with disease progression	[139]
miR-27a	100	50	CRC: 50; CA: 50;	Plasma	↑	0.746	<0.001	External validation phase results	[140]
miR-130a						0.697	<0.001		
miR-27a+miR-130a						0.801	<0.001		
miR-1539	51	49	I+II: 19; III+IV: 31; Unknown: 1;	Serum	↑	0.673	<0.003	Decreased serum expression of miR-1539 indicates LCRC	[141]

AUC: Area Under the Curve; I: CRC stage I; II: CRC stage II; IIA: CRC stage IIA; IIB: CRC stage IIB; III: CRC stage III; IIIA: CRC stage IIIA; IIIB: CRC stage IIIB; IV: CRC stage IV; N/A: information not available; CA: colorectal adenomas; T: primary tumor; N: regional lymph nodes; M: distant metastases; LCRC: left-sided CRC; ↑: upregulated expression; ↓: downregulated expression.

## Data Availability

Not applicable.

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
