# Peer review of "Clinical Applications of Liquid Biopsy in Colorectal Cancer Screening: Current Challenges and Future Perspectives"

_cells, 2022, doi:10.3390/cells11213493_

Round 1
Reviewer 1 Report
This is a well written and thorough article on a complex and novel method for early detection of colorectal cancer. The text is well balanced and combined with clear illustrations and gives an impression of an unbiased overview of the field. I am not an expert in the field myself but the over all impression is that I would recommend it for publishing
Author Response
Manuscript ID: cells-1994051
Title: “Clinical applications of liquid biopsy in colorectal cancer screening: current challenges and future perspectives” by Galos et al.
Reviewer 1 Comments and Suggestions for Authors
This is a well written and thorough article on a complex and novel method for early detection of colorectal cancer. The text is well balanced and combined with clear illustrations and gives an impression of an unbiased overview of the field. I am not an expert in the field myself but the overall impression is that I would recommend it for publishing.
We want to thank the reviewer for the time allocated to analyze our manuscript. We are pleased to know our review’s content was appreciated and we address our gratitude for reviewing our article.

Reviewer 2 Report
Thank you for giving me the opportunity to review this interesting paper:
The authors are discussion the different types of liquid biopsies in colon cancer and discussing the usage and clinical implications in addition to challenges and future direction.
My comments are as follows:
-The included biomarkers here, when talking about liquid biopsy, undergo an elimination process and have a certain half-life which is important to be discussed in a such paper and needed to be considered when evaluating circulating parameters. A.e., it was found that cfDNA clearance may be influenced by renal function.
-Percentage of cancer pts with renal impairment is not small.
-What about data on elimination which may be influenced by medication and nutritional status of pts.
- Liquid biopsy still facing many methodological challenges for routine use in clinical practice.
-Up-to-date, the isolation technology of these biomarkers may differ, and no one can say how this can impact the results and interpretation of it.
- As mentioned above it needs standardization.
- Availability and time to release of results is challenging as well.
-How liquid biopsy may influence our decision-making regarding treatment in cancer with (liquid biopsy) positive results and who are in CR. More light needed to be shed on this particular issue.
- The recently published DYNAMIC study in pts with stage II Colon cancer showed that ctDNA-based treatment decisions regarding the administration of adjuvant chemotherapy is not inferior to conventional decision making. May be to comment on that.
- There is no head-to-head comparison of commercially available assays.
- Difference in load/level of ctDNA between the different mutational status as RAS and TP53 as mentioned by author regarding the tumor mutational burden.
I wish to see more light
- on Prediction of response
- Prediction of resistance
- Monitoring of response
Thank you
Author Response
Manuscript ID: cells-1994051
Title: “Clinical applications of liquid biopsy in colorectal cancer screening: current challenges and future perspectives” by GaloÈ™ et al.
Reviewer 2 Comments and Suggestions for Authors
Thank you for giving me the opportunity to review this interesting paper:
The authors are discussion the different types of liquid biopsies in colon cancer and discussing the usage and clinical implications in addition to challenges and future direction.
We want to thank the reviewer for the time assigned to analyze our manuscript. We are confident that we will improve the current paper by answering the reviewer’s requests. We have amended our manuscript according to the reviewer’s suggestions. The changes were done using track changes. We answered each of the reviewer´s suggestion as follows:
-The included biomarkers here, when talking about liquid biopsy, undergo an elimination process and have a certain half-life which is important to be discussed in a such paper and needed to be considered when evaluating circulating parameters. A.e., it was found that cfDNA clearance may be influenced by renal function.
- Percentage of cancer pts with renal impairment is not small.
- What about data on elimination which may be influenced by medication and nutritional status of pts.
We want to thank the reviewer for the recommendations. We have amended the manuscript according to your input, therefore, in order to complete the material addressing the circulation half-life of cfDNA and CTCs, we offered more information on the elimination process of cfDNA (“The liver represents a key player in cfDNA elimination, however, the spleen, kidneys and lymphatic circulation also participate in clearance processes. Therefore, lower levels of cfDNA may be detected in plasma regardless of organ impairment [63].”). Following thorough research, to our knowledge there is no significant published data on the topic of liquid biopsy analytes and their elimination according to medication and nutritional status of patients. However, the study of literature confirmed that liquid biopsy analysis could detect indicators of cachexia in patients (“Various other conditions may lead to an increase in cfDNA levels, such as infection, systemic inflammation, cerebral infarction, acute trauma or post-transplantation, however, healthy individuals may also experience a raise in cfDNA concentrations during physical exercise [58,59]”).
- Liquid biopsy still facing many methodological challenges for routine use in clinical practice.
- Up-to-date, the isolation technology of these biomarkers may differ, and no one can say how this can impact the results and interpretation of it.
- As mentioned above it needs standardization.
- Availability and time to release of results is challenging as well.
- How liquid biopsy may influence our decision-making regarding treatment in cancer with (liquid biopsy) positive results and who are in CR. More light needed to be shed on this particular issue.
We are thankful for the suggestion to extend the topic of challenges met during liquid biopsy analysis in clinical practice. We offered additional information on the subject, attempting to successfully complete the already available knowledge regarding the limitations surrounding this novel diagnostic tool (“A critical issue implies the hemolytic process, which occurs during extraction and preparation and can influence the levels of detected miRNAs. For this reason, monitoring the hemolysis of all samples in a pre-analytical phase is mandatory [176]. Another essential constraint regards residual platelets and microparticles resulting from plasma processing that can influence miRNA measurements [177]. Moreover, it is difficult to determine if the levels of plasma miRNA are confounded by comorbidities or are cancer-related. Therefore, a significant challenge is establishing which body fluid detection method is the most appropriate for CRC screening [78].”).
- The recently published DYNAMIC study in pts with stage II Colon cancer showed that ctDNA-based treatment decisions regarding the administration of adjuvant chemotherapy is not inferior to conventional decision making. May be to comment on that.
- There is no head-to-head comparison of commercially available assays.
We thank the reviewer for these suggestions. We included in the manuscript the results of the recently published DYNAMIC II study, and additionally, we offered information about the ongoing DYNAMIC III study (“Evaluation of ctDNA levels in patients with stage II CRC has led to a decrease in adjuvant chemotherapy administration while maintaining a favorable recurrence-free survival (DYNAMIC II study) [178]. Additionally, the ongoing DYNAMIC III study attempts to evaluate a ctDNA-guided treatment approach in the setting of stage III CRC [179].”). We, the authors, feel that by addressing the purpose and outcome of these two studies, we further complete the overview on liquid biopsy and its roles in the setting of colorectal cancer. Moreover, the reader could extend their research on the topic by accessing the bibliography provided by our manuscript. Furthermore, we acknowledged the lack of published literature addressing head-to-head comparisons of currently available assays serving the analysis of liquid biopsies (“For CRC early diagnosis however, comprehensive studies comparing the available data on liquid biopsy to current screening methods are necessary in order to find the most advantageous clinical use in every-day practice.”).
- Difference in load/level of ctDNA between the different mutational status as RAS and TP53 as mentioned by author regarding the tumor mutational burden.
We appreciate the suggestion to extend our research into liquid biopsy roles in determining different mutational status. We furthered our research into the topic (“RAS assessment in mCRC is essential to select patients suitable for anti-EGFR therapy. The concordance between tissue detection and somatic mutations detected in ctDNA appeared high in patients with advanced tumors, supporting blood-based testing. Moreover, ctDNA was shown to be highly useful for monitoring treatment response. However, some clinico-pathological features, including tumor histology (mucinous) and metastatic sites (peritoneal, lung), negatively influenced RAS detection in ctDNA [158]. Along with RAS mutation, TP53 mutations were widely detected in the ctDNA of CRC patients, with a high correlation between tissue and plasma detection. In CRC patients who did not progress to metastatic disease after primary surgery, the VAF for TP53 mutations decreased. By contrast, increased levels were associated with development of liver metastasis [159]. Moreover, TP53 mutations were significantly correlated with increased VEGFA mRNA tissue expression, suggesting that these patients are expected to benefit from anti-VEGF therapy [160]. Nonetheless, TP53 mutations might occur as a consequence of several treatment strategies. In CRC, these mutations were particularly linked to cetuximab therapy, leading to resistant clones and therefore influencing treatment opportunities [161].”). By addressing this issue, we consider that we are adding valuable knowledge to this article.
I wish to see more light
- on Prediction of response
- Prediction of resistance
- Monitoring of response
We want to thank the reviewer for the suggestion of improving our manuscript. We want to emphasize that the primary purpose of our review is to offer extensive information on the roles of liquid biopsy in the setting of colorectal cancer screening, with its clinical applications, challenges and future perspectives. We feel that addressing the role of liquid biopsy in predicting response and resistance to treatment, as well as its part in monitoring response to therapy, does not represent the main objective of our work. However, we considered offering the reader an in-sight into these complex topics, by commenting on significant clinical trials and their results (“Liquid biopsy also plays an important role in assessing minimal residual disease, guiding the timing of therapeutic interventions based on ctDNA levels obtained at crucial points during treatment [180]. Furthermore, extensive research has proved that liquid biopsy offers strong indicators of early disease recurrence, with a significant median lead time of 8.7 months over conventional assessment methods [181]. Liquid biopsies have demonstrated valuable clinical application in detecting patients with acquired resistance to anti-EGFR therapy, therefore allowing a better selection of cases who may benefit from an EGFR rechallenge [182].”). We believe that the bibliography provided in this manuscript could offer important sources of information to the reader.
To conclude, we want to address our gratitude for reviewing our article.

Round 2
Reviewer 2 Report
none